# A Pharmacokinetic Dose-Optimization Study of Cabotegravir and Bictegravir in a Mouse Pregnancy Model

**DOI:** 10.3390/pharmaceutics14091761

**Published:** 2022-08-24

**Authors:** Haneesha Mohan, Kieran Atkinson, Birgit Watson, Chanson J. Brumme, Lena Serghides

**Affiliations:** 1Toronto General Hospital Research Institute, University Health Network (UHN), Toronto, ON M5G 1L7, Canada; 2British Columbia Centre for Excellence in HIV/AIDS, Vancouver, BC V6Z 1Y6, Canada; 3Division of Infectious Diseases, Department of Medicine, University of British Columbia, Vancouver, BC V5Z 1M9, Canada; 4Department of Immunology and Institute of Medical Sciences, University of Toronto, Toronto, ON M5S 1A8, Canada

**Keywords:** cabotegravir, bictegravir, HIV antiretrovirals, INSTI, pregnancy, mouse, amniotic fluid, dosing optimization, drug levels

## Abstract

Animal pregnancy models can be useful tools to study HIV antiretroviral safety and toxicity and to perform mechanistic studies that are not easily performed in humans. Utilization of clinically relevant dosing in these models improves the relevance of the findings. Cabotegravir and bictegravir are new integrase strand transfer inhibitors (INSTIs), recently approved for the treatment of people living with HIV. Studies of these drugs in pregnancy are very limited. The objective of this study was to perform a dose-optimization study of cabotegravir and bictegravir in a mouse pregnancy model with the goal of determining the dose that would yield plasma drug concentrations similar those observed in humans. Pregnant mice were administered increasing doses of cabotegravir or bictegravir in combination with emtricitabine and tenofovir by oral gavage from gestational day 11.5 to 15.5. Drug concentrations in the maternal plasma at 1 h and 24 h post drug administration and in the amniotic fluid at 1 h post drug administration were determined using high-performance liquid chromatography coupled with tandem mass spectrometry. A review of cabotegravir and bictegravir human pharmacokinetic studies are also reported. We hope these data will encourage studies of HIV antiretroviral safety/toxicity and mechanistic studies in animal pregnancy models.

## 1. Introduction

HIV integrase strand transfer inhibitors (INSTIs) are a newer class of HIV drugs that inhibit HIV integrase, a viral enzyme that mediates the transfer of virally encoded DNA into the host genome. Raltegravir (RAL) was the first INSTI to be approved by the FDA in 2007 [1,2,3], followed by elvitegravir (EVG) in 2012 [4] and dolutegravir (DTG) in 2012 [5]. In 2019, the World Health Organization (WHO) recommended DTG as preferred HIV treatment for all people living with HIV, including pregnant women, despite an early signal of a possible link between DTG and neural tube defects in infants born to women treated from conception [6]. Expanded studies reported a lower risk for neural tube defects than the initial study and the benefits of DTG, including high effectiveness, greater tolerability, greater barrier to drug resistance, and lower cost, outweighed the potential risks and led to the recommendation of DTG in the WHO 2019 guidelines.

The two newest INSTIs are bictegravir (BIC), which was approved by the FDA in 2018, and cabotegravir (CAB), which was approved in 2021 as the first long-acting injectable administered in combination with rilpivirine.

As with most medications, safety and efficacy studies in pregnancy often occur post-approval, if at all. Currently, only RAL and DTG are included in treatment guidelines for use in pregnancy. EVG is not recommended due to limited data in pregnancy, inadequate levels in the second and third trimester, and reported viral breakthroughs [7]. Both BIC and CAB are not recommended in pregnancy due to limited data.

While clinical studies are a must to determine the safety and efficacy of a specific drug in pregnancy, animal models of pregnancy can be useful tools to study the impact of antiretrovirals (ARVs) on the placenta and developing fetus and to perform mechanistic studies that are not easily performed, or even possible to do, in humans. Utilization of clinically relevant dosing in these animal models improves the relevance of the findings. We have previously published a dosing assessment of recommended ARV regimens using our established mouse pregnancy model [8,9] to determine doses of ARVs that would yield human therapeutic plasma concentrations. This previous study included dosing for DTG and RAL, in addition to multiple protease inhibitors. Here, we expand our study to include CAB and BIC, the two newest INSTIs. We begin with a review of studies of CAB and BIC in pregnancy.

### 1.1. Cabotegravir

CAB is a long-acting (LA) anti-HIV agent, formulated as 200 mg/mL CAB free acid nano-suspension, containing 200 nm crystalline nano-sized particles, manufactured by wet-bead milling process and sterilized by gamma radiation [10]. The LA effects of CAB were first investigated in Indian rhesus macaques (*Macaca mulatta*) where it showed protective efficacy against intrarectal simian/human immunodeficiency virus (SHIV) [11] and repeated high-dose intravaginal SHIV [12], providing support for the use of CAB as pre-exposure prophylaxis (PrEP) in women [13]. The efficacy of LA CAB was further investigated in pigtailed macaques, where intramuscular injections protected female macaques from repeated intravaginal SHIV [14]. In humans, LA CAB was first investigated in healthy volunteers, provided as a single-dose administration of 100–800 mg intramuscular (IM) or 100–400 mg subcutaneous (SC) injection [15,16]. An oral formulation is also available [17,18,19].

Animal pregnancy models, where CAB was administered orally to pregnant rabbits at 0, 30, 500, or 2000 mg/kg/day from gestational day 7 to 19 showed no drug-related fetal toxicities [20]. Furthermore, a study of pregnant rats treated with CAB orally at 0, 0.5, 5, or 1000 mg/kg/day from gestational day 0 to 17 reported no effects on fetal viability and no drug-related fetal malformations. A minor decrease in fetal body weight was observed at the 1000 mg/kg/day dose [20]. In a rat pre- and postnatal development study, CAB was administered orally to pregnant rats at 0, 0.5, 5, or 1000 mg/kg/day from gestational day 6 to lactation day 21. The group administered the 1000 mg/kg/day dose showed delayed onset of parturition, increased number of stillbirths, and increased number of neonatal deaths at lactation day 4, supporting an association between high CAB exposure (28 times greater than the recommended human dose) and some reproductive toxicity in animals [20].

CAB use in pregnant women living with HIV has not been evaluated, but recent findings from a ViiV-Sponsored CAB pharmacokinetics (PK) study in pregnant women living with HIV reported outcomes from 13 women [19]. Four of the 13 women received oral CAB and nine received LA CAB. Four women had live births, five women underwent elective termination, and four women experienced a miscarriage within the first nine weeks of gestation. No cases of birth defects were reported [19]. This study highlights the need for additional surveillance data and safety studies in pregnancy.

### 1.2. Bictegravir

Bictegravir (BIC) is produced as a single-tablet regimen co-formulated with emtricitabine/tenofovir alafenamide (FTC/TAF) for the treatment of adults, adolescents, children, and neonates living with HIV [21]. BIC has shown to have an improved in vitro resistance profile compared to RAL and EVG [22,23].

There is a scarcity of data on BIC used in pregnancy. In preclinical studies, BIC administered orally to pregnant rats (5, 30, or 300 mg/kg/day) and rabbits (100, 300, or 1000 mg/kg/day) from gestational day 7 through 17 and 7 through 19, respectively, was associated with no adverse embryo-fetal effects. Spontaneous abortion, decreased body weight, and maternal toxicity were observed in rabbits treated with 1000 mg/kg/day BIC [24]. In pre- and postnatal development studies, BIC administered orally to pregnant rats (up to 300 mg/kg/day) from gestational day 6 to lactation/postpartum day 24 displayed no significant adverse effects in fetuses or pups [24].

In humans, a case series evaluating the pharmacokinetics of BIC in two pregnant women showed that AUC, C_trough_, and C_max_ were 35%, 49%, and 19% lower at 33 weeks gestation compared to 6 weeks postpartum. Both participants remained virologically suppressed through delivery [25]. In an interim report, which monitored 140 pregnancies in women treated with BIC during periconception or pregnancy, three birth defects were observed [26]. Surveillance of BIC use in pregnancy continues.

## 2. Materials and Methods

### 2.1. Drugs

Bictegravir (BIC, Cat. # HY-17605/CS-0014685, MedChemExpress) and cabotegravir (CAB; Cat. # HY-15592/CS-5078, MedChemExpress) were purchased in powdered form. Both CAB and BIC were administered in combination with tenofovir disoproxil fumarate (TDF) and emtricitabine (FTC). TDF/FTC was purchased as a prescription drug under the brand name Truvada. The drug pellets were crushed to powder form. All drugs in each regimen (i.e., BIC/TDF/FTC or CAB/TDF/FTC) were mixed as per the intended combination and dose, suspended in distilled water, and sonicated for 10 min just prior to administration to the mice by oral gavage.

### 2.2. Mouse Pregnancy Model

All animal experiments were approved by the University Health Network Animal Use Committee (protocol #2575) and performed according to the policies and guidelines of the Canadian Council on Animal Care. C57BL/6J mice bred in-house (original breeders from Jackson Laboratory RRID:IMSR JAX:000664) were maintained under a 12-h light/dark cycle, with ad libitum access to food and water. Animals were acclimated to their surroundings for 1 week prior to experiment initiation. Males were isolated for 7 days prior to mating. Female mice (7–12 weeks of age) were mated with males at a ratio of 2:1. Presence of vaginal plug was denoted as gestational day 0.5 (GD0.5). Pregnancy was confirmed by weight gain of >1.5 g on GD9.5 and >3 g on GD13.5. Pregnant mice were treated with either BIC/TDF/FTC, where BIC was given at 1×, 5×, 10×, or 20× the equivalent mouse mg/kg dose, or CAB/TDF/FTC, where CAB was given at 1×, 3×, or 10× once daily by oral gavage starting on GD11.5 until GD15.5. The equivalent mouse mg/kg dose was calculated based on the human dose divided by an assumed average human weight of 60 kg. The dose of TDF/FTC was kept constant across the different dosages of BIC and CAB. On GD15.5, mice underwent saphenous bleed 24 h after the GD14.5 dose administration (trough). Immediately after the saphenous bleed, mice were dosed with the appropriate drug regimen and sacrificed 1 h post-treatment (peak), at which time blood was collected by cardiac puncture and amniotic fluid was collected by dissecting the uterus, puncturing the amniotic sac, and draining the amniotic fluid into a sterile dish. Amniotic fluid was pooled for the entire litter [9]. Samples from 2 dams per arm were analyzed for drug levels.

### 2.3. Bioanalysis of Drug Level Measurement in Blood Plasma and Amniotic Fluid Using HPLC-MSMS

BIC and CAB levels in plasma and amniotic fluid were determined using a validated, simultaneous assay using high-performance liquid chromatography coupled with tandem mass spectrometry (HPLC-MSMS). Briefly, 50 μL of plasma or amniotic fluid was spiked with internal standard (ritonavir-d6) followed by protein precipitation with acetonitrile and filtration. Filtrates were diluted 1:9 with ammonium acetate buffer (10 mM) and analyzed by the HPLC-MSMS system, consisting of a Shimadzu Nexera XR high-performance liquid chromatograph (Shimadzu, Columbia, MD, USA) coupled with a SCIEX API 4500 QqQ mass spectrometer (SCIEX, Concord, ON, Canada). Chromatographic separation by Mobile Phase A (10% methanol, 5% 50 mM ammonium acetate, 85% 0.25% acetic acid pH 3.6 ± 0.2) and Mobile Phase B (90% methanol, 5% 50 mM ammonium acetate, 5% 0.25% acetic acid pH 6.6 ± 0.2) was performed under linear gradient conditions (flow rate 0.7 mL/min) on a reverse-phase Zorbax XDB-C18 column (Agilent Technologies, Santa Clara, CA, USA) maintained at 40 °C. Analytes were detected in positive electrospray ionization mode with selected multiple-reaction monitoring (MRM), and the transitions were *m*/*z* 450.1/261.1, 406.1/107.1, and 727.3/146.1 for BIC, CAB, and the internal standard, respectively. HPLC-MSMS data were acquired using Analyst (v 1.6.3, SCIEX, Concord, ON, Canada) and analyzed with MultiQuant software (v 3.0.3, SCIEX, Concord, ON, Canada). Matrix effects were evaluated using blank mouse plasma spiked with BIC and CAB (recovery with plasma: 77–81% and 75–93% for BIC and CAB, respectively over low, medium, and high concentration). No peaks were detected in blank mouse plasma at the retention time with the specific ion settings for each compound. The method was linear over the concentration range of 89.6–20,000 ng/mL for BIC and CAB (r^2^ ≥ 0.998). Positive controls spiked in-house and prepared by an external laboratory during a sample exchange program were included in all runs. Measured concentration of controls was 93–117% and 89–102% of the nominal values for BIC and CAB, respectively, in internal controls, and 80–104% and 76–92% for BIC and CAB, respectively, in external controls.

## 3. Results

The pharmacokinetics estimates of CAB and BIC in the plasma and amniotic fluid are shown in Table 1. The daily recommended human dose for CAB is 30 mg per day and for BIC is 50 mg per day, and the equivalent 1× mouse daily dose is 0.5 mg/kg for CAB and 0.83 mg/kg for BIC. CAB and BIC were co-administered with 50 mg/kg TDF and 33.3 mg/kg FTC. The dose of TDF/FTC was kept the same across all experiments.

To determine the optimum dosage that would yield human therapeutic levels in pregnant mice, we administered CAB at 1×, 3×, 10×, and BIC at 1×, 5×, 10×, and 20× mouse doses. Mice were treated once daily by gavage starting on GD11.5 of pregnancy until GD15.5. Plasma levels of CAB and BIC were assessed at 1 h (peak) and 24 h (trough) post drug administration. Amniotic fluid levels of CAB and BIC were assessed at 1 h post drug administration. The resulting plasma and amniotic concentrations for CAB and BIC are shown in Table 1.

As a comparison, we provide a summary of human pharmacokinetic studies of CAB in Table 2 and BIC in Table 3. The tables show dosing regimen and associated peak (C_max_) and trough (C_min_) plasma concentrations from studies performed in participants (male and female) with and without HIV, and in pregnant women living with HIV.

### 3.1. Plasma Cmax Concentrations

Therapeutically relevant concentrations (660 ng/mL for CAB [16] and 162 ng/mL for BIC [27]) were achieved for both CAB and BIC at all concentrations tested at 1 h post dosing. C_max_ concentrations for CAB were below those reported in human studies for the 1× and 3× dose but approximated human C_max_ values at the 10× dose. For BIC, C_max_ concentrations were below those reported for the 1× dose, above those reported for the 10× and 20× dose but approximated those reported in human studies for the × dose.

**Table 2 pharmaceutics-14-01761-t002:** Literature review of HIV CAB pharmacokinetic studies in humans.

Drug	Co-Administered Drug(s) and Dose	Participant Characteristics	CAB Dosage, Frequency, and Route		Human Plasma Levels (ng/mL)		Reference
					C_max_	C_min_	
**CAB**	**-**	Healthy Participants	Single oral dose	5 mg	1150	420	[17]
				10 mg	2150	830	
				25 mg	4380	1930	
				50 mg	8570	3820	
			Repeat oral dose	5 mg	2080	1060	
				10 mg	4290	2390	
				25 mg	9470	5400	
		Participants with HIV	Single oral dose	5 mg	520	230	
				30 mg	2840	1250	
			Repeat oral dose	5 mg	1020	-	
				30 mg	6370	3280	
	-	Healthy Participants	CAB given as IM (gluteal) injection	100 mg IM	200	-	[15]
				200 mg IM	300	-	
				400 mg IM	700	-	
				400 mg (200 mg × 2)	1400	-	
				800 mg IM (400 mg × 2)	3300	-	
			CAB given as SC (abdominal) injection	100 mg	200	-	
				200 mg	500	-	
				400 mg (200 mg × 2)	900	-	
	Rilpivirine LA 300 mg/mL suspension given as IM (gluteal) injection monthly	Healthy Participants	Oral lead-in followed by LA CAB given as IM (gluteal) or SC (abdominal) injection	30 mg	8300	4900	[16]
				200 mg SC	2100	1660	
				200 mg IM	2200	1610	
				400 mg IM	4400	3270	
				800 mg IM	3300	1100	
		Healthy Participants (Males)	30 mg oral phase, 800 mg LA CAB repeated dose every 12 weeks	800 mg LA CAB:			[28]
				1st injection	4260	302	
				2nd injection	5220	331	
				3rd injection	4910	387	
	-	Healthy Participants (Males)	30 mg oral phase, 800 mg LA CAB repeated dose every 12 weeks	800 mg LA CAB:			[29]
				1st injection	2670	490	
				2nd injection	2570	780	
				3rd injection	3390	820	
				4th injection	-	-	
				5th injection	-	-	
		Healthy Participants (Females)	30 mg oral phase, 800 mg LA CAB repeated dose every 12 weeks	800 mg LA CAB:			
				1st injection	1890	950	
				2nd injection	2290	1350	
				3rd injection	3010	1650	
				4th injection	-	-	
				5th injection	-	-	
		Healthy Participants (Males)	30 mg oral phase, 600 mg LA CAB repeated dose every 12 weeks	600 mg LA CAB:			
				1st injection	2510	1790	
				2nd injection	3900	1290	
				3rd injection	2960	1110	
				4th injection	2960	1460	
				5th injection	3820	1680	
		Healthy Participants (Females)	30 mg oral phase, 600 mg LA CAB repeated dose every 12 weeks	600 mg LA CAB:			
				1st injection	1580	1330	
				2nd injection	2960	1820	
				3rd injection	3460	2040	
				4th injection	3330	2060	
				5th injection	3660	2030	
	CAB/rilpivirin LA injectable every 4 weeks	ART-naïve, Switch study; ART- experienced, virologically suppressed	Oral	30 mg once daily	8000	4600	[30,31]
			Initial injection	600 mg IM initial dose	8000	1500	
			Monthly injection	400 mg IM monthly	4200	2800	
	CAB/rilpivirine LA injectable every 4 or 8 weeks	Participants with HIV	Repeat-dose every 4 week	400 mg		2740	[32]
		Participants with HIV	Repeat-dose every 8 week	600 mg		1670	

CAB, cabotegravir; LA, long-acting; SC, sub-cutaneous; IM, intramuscular.

**Table 3 pharmaceutics-14-01761-t003:** Literature review of HIV BIC pharmacokinetic studies in humans.

Drug	Co-Administered Drug(s) and Dose	Participant Characteristics	BIC Dosage, Frequency, and Route		Human Plasma Levels (ng/mL)		Reference
					C_max_	C_min_	
**BIC**	-	Participants with HIV	Multiple-dose	5 mg	741.5	225.3	[27]
				25 mg	3475	1052.3	
				50 mg	6080	2053	
				100 mg	12,235	4520	
	Coformulated BIC 50 mg, FTC 200 mg, TAF 25 mg	Participants with HIV	Oral	50 mg	-	2310	[33]
	Coformulated BIC 50 mg, FTC 200 mg, TAF 25 mg	Participants with HIV	Oral	50 mg	-	2576	[34]
	Coformulated BIC 50 mg, FTC 200 mg, TAF 25 mg	Participants with HIV	Oral	50 mg	-	2282.9	[35]
	Coformulated BIC 50 mg, FTC 200 mg, TAF 25 mg	Participants with HIV	Oral	50 mg	-	2038.2	[36]
	-	Healthy Participants	Oral	30 mg fasting	3450	1660	[37]
				30 mg fed	3950	1930	
	Coformulated BIC 50 mg, FTC 200 mg, TAF 25 mg	Pregnant Woman 1 GW 33	Oral	50 mg	3820	630	[25]
		Pregnant Woman 2 GW 33			4840	-	

BIC, bictegravir; TAF, tenofovir alafenamide; FTC, emtricitabine; GW, gestational week.

### 3.2. Plasma Cmin Concentrations

Both CAB and BIC were detectable in plasma 24 h post dosing for all doses tested; however, for the 1× BIC, the value was below the lowest value of the linear portion of the standard curve (i.e., 89.6 ng/mL). The C_min_ values for both CAB and BIC were below those reported in human studies with the exception of the highest doses tested, 10× for CAB and 20× for BIC.

### 3.3. Amniotic Fluid Concentrations

Amniotic fluid was collected 1 h after the last administered dose. Both CAB and BIC were detectable in the amniotic fluid at all concentrations tested, although levels in the 1× and 3× CAB and in the 1× BIC were below the linear range of the standard curve. The ratio of drug in the amniotic fluid to that in maternal plasma was low but consistent across doses, ranging from 3.4–5.5% for 1× to 10× CAB, and 2.1–3.6% for 1× to 10× BIC. At the 20× BIC dose, the ratio was higher at 11.2%.

## 4. Discussion

Our objective with this study was to provide more data on the pharmacokinetic dosing-optimization of ARVs in a mouse pregnancy model, expanding on our previous study [9] to include the two newest INSTIs, CAB and BIC. Our goal is to inform and encourage the use of mouse models for the study of ARVs in pregnancy. To ensure clinically relevant dosing in animal models, drug optimization is needed, as adjusting the human dose for body weight and metabolic rate seldom yields drug concentrations similar to those in humans. CAB and BIC are not recommended for use in pregnancy due to lack of data, so more investigation of these drugs in pregnancy are needed.

Based on our studies, a 10× dose of CAB (i.e., 5 mg/kg/day) resulted in a C_max_ concentration of ~3500 ng/mL and a C_min_ of ~1300 ng/mL, which are similar to those published in human pharmacokinetic studies. Spreen et al. [17], reported a C_max_ of 2840 ng/mL and C_tau_ of 1250 ng/mL in participants with HIV treated with a 30 mg oral dose of CAB. Our observed C_max_ and C_min_ for the 10× dose of CAB are also similar to those reported for the injectable formulations of CAB [29] and within the range of values reported for the injectable CAB co-formulated with rilpivirine, being somewhat higher than one study and lower than a second study [31,32]. Concentrations of CAB in pregnancy are likely to differ to some degree from those in non-pregnant individuals; however, no pharmacokinetic studies of CAB in pregnancy have been reported at this time.

C_max_ concentrations of 6080 ng/mL and C_tau_ of 2053 ng/mL have been reported in persons living with HIV administered a 50 mg daily dose of BIC [27]. We observed a similar C_max_ value of ~7700 ng/mL with a 5× dose of BIC (i.e., 4.15 mg/kg/day) in our model. However, C_min_ values for BIC were lower than those reported in humans for all doses tested except the 20× dose, although the 20× dose resulted in a C_max_ value that was much higher than that seen in humans. As such, we would recommend a dose of 5× BIC, as the most clinically relevant dose.

A single study reported pharmacokinetic profiles of BIC administered with FTC/TAF in two pregnant women at gestational week 33 and 6-weeks postpartum. The first woman had a C_max_ of 3830 ng/mL and a C_min_ of 630 ng/mL at 33 weeks gestation, a 49% and 19% decrease, respectively, from the values seen at 6-weeks postpartum, but still within the therapeutic range. A C_max_ of 4840 ng/mL was reported for the second woman at 33 weeks gestation, with no postpartum data available [25]. Pregnancy is associated with physiological changes that can impact absorption, distribution, protein binding, metabolism, and clearance of drugs, often resulting in lower drug concentrations in the third trimester than in non-pregnancy [38]. More PK studies in pregnancy are warranted for both BIC and CAB.

There are minimal data of maternal-to-fetal transfer of CAB and BIC. BIC concentrations in maternal and cord blood at delivery were measured in the two pregnant women mentioned above [25]. In the first woman, cord blood concentrations for BIC were 1350 ng/mL and maternal blood concentrations were 910 ng/mL 20 h after drug intake, resulting in a cord blood to maternal blood ratio of 1.49. For the second woman, cord and maternal blood concentration were 2910 ng/mL and 2060 ng/mL, respectively, 7 h after drug intake, resulting in a cord to maternal blood ratio of 1.42.

Our data show that amniotic fluid drug concentrations 1 h after dosing were 112.4 ng/mL for 10× CAB and 142 ng/mL for 5× BIC, both below the reported therapeutic concentrations. The amniotic fluid to maternal plasma ratio was between 3.4% to 5.5% for 1× to 10× CAB and 2.1% to 3.6% for 1× to 10× BIC. A study on placental transfer of CAB and BIC in an ex-vivo dually perfused human cotyledon model reported low placental transfer for both CAB and BIC [39]. They reported concentrations in the maternal and fetal compartments of 550 ng/mL and 48 ng/mL, respectively, for CAB, and 1650 ng/mL and 126 ng/mL, respectively, for BIC, resulting in a maternal-to-fetal ratio of 10% for CAB and 7% for BIC.

Our study has some limitations. CAB and BIC were administered with TDF/FTC. Currently, CAB is approved to be co-administered with rilpivirine, while BIC is available as a single pill co-formulated with TAF/FTC. Differences in the backbone regimen may lead to differential drug–drug interaction that could potentially influence CAB and BIC levels, although these are unlikely [40]. Further, we administered all our drugs by oral gavage, while CAB is most likely to be administered as an intramuscular injection, although an oral formulation has been approved. Drug–drug interactions are likely to differ between the oral and intramuscular route of administration. Intramuscular administration allows for absorption of drugs directly into the bloodstream, bypassing first-pass metabolism (stomach/intestine), which is likely to provide some protection against drug–drug interactions. However, studies directly addressing this issue in the context of HIV antiretrovirals are lacking [41].

In summary, we report a dosing evaluation of CAB and BIC regimens (administered with TDF/FTC) in mouse plasma and amniotic fluid using a mouse pregnancy model. We hope to encourage the use of mouse models for mechanistic studies of ARV safety and toxicity, including ARV effects on fetal and placental development.

## Figures and Tables

**Table 1 pharmaceutics-14-01761-t001:** Pharmacokinetic dosing evaluation of cabotegravir and bictegravir in a mouse model.

Drug	Human Dosing Regimen	Equivalent Mouse Dose (1×)	Daily Mouse Dose Tested	Co-Admin with	Plasma Concentration (ng/mL) *		Amniotic Fluid (ng/mL) *	Amniotic: Maternal Plasma Ratio *
					1 h (C_max_)	24 h (C_min_)	1 h	1 h
**CAB**	30 mg	0.5 mg/kg/day	0.5 mg/kg (1×)	TDF 50 mg/kg + FTC 33.3 mg/kg	912.8 (93.7)	260.1 (15.5)	44.9 (15.4)	0.050 (0.022)
			1.5 mg/kg (3×)		1269 (574)	480.5 (88.2)	59.9 (11.0)	0.055 (0.033)
			5 mg/kg (10×)		3583 (1983)	1306 (402)	112.4 (33.6)	0.034 (0.009)
**BIC**	50 mg	0.83 mg/kg/day	0.83 mg/kg (1×)	TDF 50 mg/kg + FTC 33.3 mg/kg	1758 (3.6)	64.0 (43.3)	36.9 (3.5)	0.021 (0.002)
			4.15 mg/kg (5×)		7713 (2300)	365.5 (104.9)	142 (80.7)	0.021(0.017)
			8.3 mg/kg (10×)		12,789 (17.3)	765.1 (3.4)	462.8 (160.6)	0.036 (0.013)
			16.6 mg/kg (20×)		14,749 (4456)	2215 (752)	1728 (966)	0.112 (0.031)

* Data are presented as mean (SD) (*n* = 2/arm). CAB, cabotegravir; BIC, bictegravir; TDF, tenofovir; FTC, emtricitabine.

## Data Availability

Not applicable.

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
