# Peer review of "A Pharmacokinetic Dose-Optimization Study of Cabotegravir and Bictegravir in a Mouse Pregnancy Model"

_pharmaceutics, 2022, doi:10.3390/pharmaceutics14091761_

Round 1

Reviewer 1 Report

HIV integrase strand transfer inhibitors are a newer class of HIV drugs that inhibit HIV integrase, a viral enzyme that mediates the transfer of virally encoded DNA into the host genome. The authors highlight the fact that dolutegravir (DTG) is an integrase strand transfer inhibitor that was approved by the FDA in 2012 and in 2019 the World Health Organization recommended DTG as a preferred HIV treatment for all people living with HIV, including pregnant women. This was despite an early signal of a possible link between DTG and neural tube defects in infants born to women treated from conception.

More recent studies working with DTG reported a lower risk for neural tube defects than the initial study and demonstrated the benefits of DTG, including high effectiveness, greater tolerability, greater barrier to drug resistance, and lower cost. The recent studies suggested that the potential risks were outweighed by the benefits and this led to the recommendation of DTG in the WHO 2019 guidelines.

The two newest integrase strand transfer inhibitors are bictegravir (BIC) which was approved by the FDA in 2018, and cabotegravir (CAB) which was approved in 2021 as the first long-acting injectable drug administered in combination with rilpivirine. Bictegravir and cabotegravir are currently not recommended in pregnancy due to limited data.

In this article Mohan et al wanted to perform a dose-optimization study of two new integrase strand transfer inhibitors cabotegravir and bictegravir in a mouse pregnancy model with the aim of determining which dose would yield plasma drug concentrations similar to those currently observed in humans. Cabotegravir and bictegravir have recently been approved for the treatment of people living with HIV and so are of particular interest. The authors took pregnant C57BL/6J mice and administered increasing doses of cabotegravir or bictegravir by oral gavage from gestational day 11.5 to 15.5 in combination with emtricitabine and tenofovir. The authors used high-performance liquid chromatography along with tandem mass spectrometry to determine drug concentrations in the maternal plasma at 1 hour and 24 hours post drug administration as well as in the amniotic fluid at 1 hour post drug administration. Mohan et al also wanted to present a review of cabotegravir and bictegravir human pharmacokinetic studies in this article. The authors wish to encourage more experimental work to assess HIV antiretroviral safety, toxicity and mechanistic studies in animal pregnancy models.

The authors begin their article with a review of both bictegravir and cabotegravir in pregnancy models including the use of cabotegravir as pre-exposure prophylaxis in women. The authors note that cabotegravir use in pregnant women living with HIV has not been evaluated to date, but a conference paper highlights the need for additional surveillance data and safety studies in pregnancy. The authors then continue to say that there is a scarcity of data on bictegravir being used in pregnancy but that surveillance of bictegravir use in pregnancy continues.

The authors conclude that their article reports a dosing evaluation of cabotegravir and bictegravir regimens (administered in conjunction with tenofovir and emtricitabine) in mouse plasma and amniotic fluid using a mouse pregnancy model. They state that they want to encourage the use of mouse models for mechanistic studies of antiretroviral safety and toxicity, including antiretroviral effects on fetal and placental development.

Main points and comments:

1.     Can the authors explain how they have recovery percentages above 100% on lines 172 and 173?

2.     Table 1 seems to be a little bit squashed (such as “Equivalent Mouse dose (1x)” and the mg/kg/day numbers under this heading. Can this be adjusted in any way to make it slightly clearer?

3.     Also, with regards to Table 1, I am slightly concerned about the phrase “…. (n=2-4).” Can the authors be more specific as there is no indication which groups were n=2 and which were n=3 and which were n=4. It would have been nice to have n=3 across the board as opposed to varying numbers. Can the authors please make it clear how many animals were used at each time point and in each group? There is no mention of this in the Materials and Methods (unless I have missed it).

4.     Table 2 is a useful comparison for Table 1 although once again, some of the headings and columns are a little bit squashed. Can this be sorted out to make it less cluttered?

5.     A really minor point but it would be good to standardise….there appears to be an extra full stop or an extra space or a lack of a space in some of the headings (such as lines 117, 118, 127, 232). Line 98 looks correct and the others seem to have a variation on a theme. Can these all be made consistent?

6.     The authors have highlighted a very important point that more pharmacokinetic studies in pregnancy are needed for both cabotegravir and bictegravir in order to allow better treatment for women. They have also demonstrated that there are minimal data available relating to maternal to fetal transfer of cabotegravir and bictegravir.

7.     The authors have acknowledged themselves that there are a few limitations to the studies they are presenting in this article and I have slight concerns about some of these…in particular the co-administration of cabotegravir and bictegravir with tenofovir and emtricitabine. As the authors state, “currently CAB is approved to be co-administered with rilpivirine, while BIC is available as a single pill co-formulated with TAF/FTC”. Can the authors please comment fully on this?

8.     I appreciate there may be differential drug-drug interactions that could potentially influence cabotegravir and bictegravir levels so can the authors please comment on the differences in the backbone regimen.

9.     I think the studies presented in this article are useful and do provide a foundation to continue further work in an under-represented area of research.

10.  The authors state that they administered all their drugs by oral gavage in their present study although cabotegravir is most likely to be administered as an intramuscular injection (even though an oral formulation has been approved). Can the authors clarify if it is possible for them to work with an intramuscular model rather than oral gavage? Is bioavailability much better via the oral route as opposed to the intramuscular route? Why did the authors choose oral gavage rather than intramuscular?

Author Response

We thank the reviewer for their thorough review and their very helpful comments.  We hope that we have addressed them adequately.

Main points and comments:

  1. Can the authors explain how they have recovery percentages above 100% on lines 172 and 173?

We thank the reviewer for the question. The term “recovery” means the measured concentration expressed as a percentage of the expected value.  "Recovery" can therefore be >100% for a number of reasons including matrix effects, data analysis artefacts, or something as simple as minor pipetting inaccuracies. Assay validation guidelines typically state that "Recovery" should be ±15% of the nominal (theoretical) value (or ±20% of the nominal value for concentrations around the LLOQ). To be explicit in explanation, we have rephrased the line 172 and 173, which now states, “Measured concentration of controls was 93-117% and 89-102% of the nominal values for BIC and CAB respectively in internal controls, and 80-104% and 76-92% for BIC and CAB respectively in external controls.”

  1. Table 1 seems to be a little bit squashed (such as “Equivalent Mouse dose (1x)” and the mg/kg/day numbers under this heading. Can this be adjusted in any way to make it slightly clearer?

We thank the reviewer for noticing the misalignment in Table 1. The table margins are adjusted and the heading for column three now clearly states, “Equivalent Mouse dose (1x)” and “mg/kg/day”

  1. Also, with regards to Table 1, I am slightly concerned about the phrase “…. (n=2-4).” Can the authors be more specific as there is no indication which groups were n=2 and which were n=3 and which were n=4. It would have been nice to have n=3 across the board as opposed to varying numbers. Can the authors please make it clear how many animals were used at each time point and in each group? There is no mention of this in the Materials and Methods (unless I have missed it).

Thank you for the comment.  We have revised our methods section (mouse pregnancy model) to clearly state the number of animals used in the experiments.  We have also clarified this in Table 1.

  1. Table 2 is a useful comparison for Table 1 although once again, some of the headings and columns are a little bit squashed. Can this be sorted out to make it less cluttered?

We thank the reviewer for the comment.  We have reformatted the table and we hope this is now clearer for the reader.

  1. A really minor point but it would be good to standardise….there appears to be an extra full stop or an extra space or a lack of a space in some of the headings (such as lines 117, 118, 127, 232). Line 98 looks correct and the others seem to have a variation on a theme. Can these all be made consistent?

We thank the reviewer for noticing this error. We have removed the extra full stop and added space/removed space to make all the headings look consistent.

  1. The authors have highlighted a very important point that more pharmacokinetic studies in pregnancy are needed for both cabotegravir and bictegravir in order to allow better treatment for women. They have also demonstrated that there are minimal data available relating to maternal to fetal transfer of cabotegravir and bictegravir.

We thank the reviewer for their comment.

  1. The authors have acknowledged themselves that there are a few limitations to the studies they are presenting in this article and I have slight concerns about some of these…in particular the co-administration of cabotegravir and bictegravir with tenofovir and emtricitabine. As the authors state, “currently CAB is approved to be co-administered with rilpivirine, while BIC is available as a single pill co-formulated with TAF/FTC”. Can the authors please comment fully on this?

This is a limitation of our study, as mentioned in our discussion.  However, no differences are expected between TAF and TDF and BIC administration, and this has been added to the discussion.  In general, no interactions have been observed between integrase strand transfer inhibitors in general and TDF.  Interaction between CAB and rilpivirine are also not expected and have not been reported to our knowledge.  However, specific studies will need to be done to address these questions in the mouse model, and this is something we will pursue in future studies.  Additional discussion has been added to the limitations section of the discussion.

  1. I appreciate there may be differential drug-drug interactions that could potentially influence cabotegravir and bictegravir levels so can the authors please comment on the differences in the backbone regimen.

Modeling studies have suggested that drug-drug interactions could be an issue between bictegravir and drugs that strong inhibitors or inducers of CYP3A and UGT1A1 – like atazanavir (Stader F et al. Clin Pharmacol Ther 2021). Neither TDF or TAF interact with CYP3A4 or UGT1A1, and so no interactions are expected between BIC and TDF or TAF. Cabotegravir is also metabolized mainly by UGT1A1, so potential interactions can occur with strong inducers or inhibitors of UGT1A1.  No interactions have been reported between CAB and TDF or TAF. Rilpivirine is metabolized mainly by CYP3A4.

  1. I think the studies presented in this article are useful and do provide a foundation to continue further work in an under-represented area of research.

We thank the reviewers for their comment and appreciate their encouraging feedback.

  1. The authors state that they administered all their drugs by oral gavage in their present study although cabotegravir is most likely to be administered as an intramuscular injection (even though an oral formulation has been approved). Can the authors clarify if it is possible for them to work with an intramuscular model rather than oral gavage? Is bioavailability much better via the oral route as opposed to the intramuscular route? Why did the authors choose oral gavage rather than intramuscular?

We chose to administered CAB by oral gavage rather than intramuscularly so as to keep consistent with the methodology of our model.  However, we agree with the reviewer that intramuscular administration should be explored and we aim to do this in future experiments.  It is possible that the bioavailability of the drug will vary depending on the route of administration, as intramuscular administration bypasses first-pass metabolism, so a dosing optimization experiment will need to be repeated using the intramuscular route. However, CAB is primarily metabolised by UGT1A1 and not CYP3A4, so bypassing the stomach/intestine may not be as consequential for CAB.  It’s worth noting that we have been able to achieve an oral dosing of CAB that yields human relevant drug plasma levels similar to those seen with intramuscular administration of CAB. Additional discussion was added to the revised manuscript.  See lines 363-367 of revised manuscript.

Reviewer 2 Report

Here are some suggestions for this paper   #1. Line 135, please provide the unit of GD 0.5, Do you mean 0.5 (day) ? #2. Line 178 , 1X , Do you mean Equivalent daily dose? #3. Line 287, the authors reported the drug level via oral use of CAB and CAB is more likely to be used by intra-muscular route. Is there difference in previous reported literature regarding the difference of these two different routes ?   

Author Response

We thank the reviewer for their helpful comments, and we hope we have addressed them adequately.

Here are some suggestions for this paper  

#1. Line 135, please provide the unit of GD 0.5, Do you mean 0.5 (day)?

GD0.5 refers to gestational day 0.5.  We have clarified this in the methods section where the term GD is first used.

 #2. Line 178, 1X , Do you mean Equivalent daily dose?

We apologize for the confusion.  We have revised our methods section to clearly state that the 1X dose refers to the equivalent mouse mg/kg dose, which is calculated based on the human dose divided by an assumed average human weight of 60kg.

 #3. Line 287, the authors reported the drug level via oral use of CAB and  CAB is more likely to be used by intra-muscular route. Is there difference in previous reported literature regarding the difference of these two different routes?   

No mouse data are available for CAB administered by intramuscular injection. The available human data are listed in Table 2.  While we expect drug levels to differ based on route of administration, we have selected an oral dose of CAB that yields plasma levels equivalent to those seen with intramuscular studies of long-acting formulation, as these are the drug levels that would be clinically relevant.  This is discussed in paragraph 2 of the discussion.  Please also see the response to reviewer 1 – question #10.  Additional discussion was added to the revised manuscript to address this issue further.  See lines 363-367 of revised manuscript.